# Targeting Bacterial Nanocellulose Properties through Tailored Downstream Techniques

**DOI:** 10.3390/polym16050678

**Published:** 2024-03-02

**Authors:** Everton Henrique Da Silva Pereira, Marija Mojicevic, Cuneyt Erdinc Tas, Eduardo Lanzagorta Garcia, Margaret Brennan Fournet

**Affiliations:** PRISM Research Institute, Technological University of the Shannon: Midlands Midwest, Dublin Rd, Co. Westmeath, N37 HD68 Athlone, Ireland; pereiraehs@gmail.com (E.H.D.S.P.); marija.mojicevic@tus.ie (M.M.); cuneyt.tas@tus.ie (C.E.T.); margaret.brennanfournet@tus.ie (M.B.F.)

**Keywords:** bacterial nanocellulose, biopolymers, downstream, materials, treatment, Komagateibacter, membranes

## Abstract

Bacterial nanocellulose (BNC) is a biodegradable polysaccharide with unique properties that make it an attractive material for various industrial applications. This study focuses on the strain *Komagataeibacter medellinensis* ID13488, a strain with the ability to produce high yields of BNC under acidic growth conditions and a promising candidate to use for industrial production of BNC. We conducted a comprehensive investigation into the effects of downstream treatments on the structural and mechanical characteristics of BNC. When compared to alkaline-treated BNC, autoclave-treated BNC exhibited around 78% superior flexibility in average, while it displayed nearly 40% lower stiffness on average. An SEM analysis revealed distinct surface characteristics, indicating differences in cellulose chain compaction. FTIR spectra demonstrated increased hydrogen bonding with prolonged interaction time with alkaline solutions. A thermal analysis showed enhanced thermal stability in alkaline-treated BNC, withstanding temperatures of nearly 300 °C before commencing degradation, compared to autoclaved BNC which starts degradation around 200 °C. These findings provide valuable insights for tailoring BNC properties for specific applications, particularly in industries requiring high purity and specific mechanical characteristics.

## 1. Introduction

Bacterial nanocellulose (BNC) is a biodegradable and naturally occurring polysaccharide produced by diverse species of bacteria, belonging to different genera, such as *Komagataeibacter*, *Acetobacter*, *Achrobacter*, and *Azotobacter*, among many others [1,2]. The main difference between BNC and plant-produced cellulose (PC) is the purity, wherein the absence of hemicellulose, lignin, and pectin in BNC increases its biocompatibility and significantly simplifies the purification process [3]. Furthermore, several features, including high crystallinity, mechanical strength, water retention, moldability, thermal stability, and more, suggest that BNC often exhibits superior characteristics compared to plant cellulose, particularly in applications within the food and biomedical fields where the highest purity is preferred [4,5]. Despite the numerous benefits of BNC, its production still faces challenges in terms of cost-efficiency, related mainly to the high cost of the production media (around 30% of total cost) [6,7]. Current research efforts are focused on developing more efficient strains, optimizing cost-effective feedstock media, and streamlining downstream processes [8,9,10,11,12].

The BNC obtained post-fermentation generally contains impurities, including cells and medium components. A subsequent downstream treatment is employed in order to purify the final product from any remaining bacteria within the BNC matrix, as a step preceding the drying process [13,14]. Several purification methods can be employed, such as extended soaking of BNC in water [6], swift washes with water followed by an alkaline treatment with NaOH [15], or by incorporating heat (60–80 °C) along with the washes [16]. Several concentrations of NaOH are used from laboratory to industrial scale, ranging from 0.5 M, adding heat, for disinfection [17,18,19], to much higher concentrations for sterilization levels [20]. However, the interaction with alkaline solution could impact nanofibril structure and the mechanical properties of BNC [21,22], turning it brittle [23,24], which could limit its application. Therefore, it is important to explore the effects of different downstream treatments and their effects on the mechanical properties in order to fine-tune the process according to the desired properties for specific applications. For instance, a method that allows the BNC to retain a higher flexibility could be preferable for applications such as scaffolds for bone and tissue engineering, medical implants [25], textile applications [26], and electrical and electronic industries [27], among others [28].

Nevertheless, in relation to strains, the strain (formerly *Gluconacetobacter medellinensis* ID13488) has demonstrated the ability to produce crystalline BNC under highly acidic growth conditions (pH 3.55) [29]. This makes it a promising candidate for recycling acid residues and for use in other industrial acid-related applications [29,30].

Given the potential of the strain *K. medellinensis* ID13488 to function as a BNC producer under industrially relevant and environmentally sustainable conditions, it is important to conduct a more comprehensive investigation into the properties of the material derived from this species and how downstream treatments may influence its application. To address this, our study involved the characterization of alkaline and autoclave treated BNC samples produced by this strain, in order to explore and compare the effects of such treatments on the structural and mechanical characteristics of the materials.

## 2. Materials and Methods

### 2.1. Microorganisms and Culture Condition

The *Komagataeibacter medellinensis* (ID13488) strain was stored at −80 °C in a Hestrin-Schramm medium (HS) stock modified with 25% glycerol broth to maintain consistency across experiments. The HS medium consisted of glucose (2 g/L), peptone (0.5 g/L), yeast extract (0.5 g/L), disodium phosphate (0.27 g/L), and citric acid (0.15 g/L). Aliquots were taken from this stock and subjected to the appropriate propagation process. To ensure robust growth and consistent results, all precultures were conducted under dynamic conditions (180 rpm) at a temperature of 30 °C for 48 h in a liquid HS medium. *K. medellinensis* cultivation was carried out in 15 Erlenmeyer flasks, each with a volume of 250 mL. The flasks contained 10% *v/v* of *K. medellinensis* preculture within 100 mL of HS broth and were kept in a static condition at 30 °C for a 10-day cultivation period. Furthermore, five randomly selected group samples in triplicate were applied for further downstream processing.

### 2.2. Downstream Processing

#### 2.2.1. Effect of KOH and NaOH Concentrations

In order to investigate the effect of KOH and NaOH concentrations on the mass difference in the BNC dry membranes were harvested from 100 mL cultures (from different batches) and underwent downstream processing. Therefore, they were immersed in KOH or NaOH solutions (0.2 to 1.0 M) for 60 min at 100 °C. After treatment, the membranes were rinsed with deionized water until the pH reached neutrality and then dried under room temperature for 48–72 h, for further mass determination.

#### 2.2.2. Alkaline and Autoclave treatments

BNC membranes were harvested from 100 mL cultures and underwent different treatments for downstream processing as described in Table 1.

The first treatment involved immersing membrane samples in 0.1 M NaOH or 0.1 M KOH at 100 °C for 60 min. Subsequently, samples were thoroughly rinsed with deionized water until the pH reached neutrality. Finally, the membranes were gently air-dried using paper and left at room temperature for 48–72 h (dried to constant weight). Since the exposure to the alkaline solution was limited to approximately 1 h, this treatment was labeled “short exposure treatment” (SE).

The second treatment regimen entailed submerging membrane samples in alkaline solutions as previously described. Neutralization was achieved by immersing the membranes in water overnight. Following neutralization, the membranes were gently air-dried using paper and left at room temperature for 48–72 h. As exposure to the alkaline solution was extended to 12 h contact, this treatment was designated “long exposure treatment” (LE).

The final treatment involved immersing the membranes in deionized water and disinfecting them using 120 °C vapor steam during 20 min in a Raypa Steam Sterilizer AE-75-DRY (Raypa, Barcelona, Spain) to kill the bacteria remnants from the film structure [31]. After disinfection, the membranes were air-dried at room temperature for 48–72 h until their dried mass could be accurately measured.

### 2.3. BNC Weight Variation and Thickness Determination

The dry BNC membranes were measured using a 10^−3^ readability scale (Chyo Balance Corp., Kyoto, Japan) following the chemical and thermal downstream processes. In addition, the thickness of the BNC samples was measured using a digital micrometer. Five measurements through the length were taken for each sample, and the average value was calculated.

### 2.4. Scanning Electron Microscopy (SEM) Imaging and Analysis

The surface of the samples was analyzed using scanning electron microscopy (SEM). Back-scattered electron mode images were captured using the Mira XMU SEM (Tescan™, Brno, Czech Republic) with an accelerating voltage of 9 kV. To prepare the samples for SEM imaging, they were placed on an aluminum stub and coated with a thin layer of gold using Baltec SCD 005 sputter coater (Bal-Tec, Coesfeld, Germany). The sputtering process lasted 110 s and was performed under a vacuum pressure of 0.1 mbar. This coating process was carried out prior to the imaging analysis to ensure optimal sample visualization.

### 2.5. Dynamics Mechanical Analysis

The tensile behavior analyses were performed on a DMA TA Q800 V21.2 Build 73 instrument (TA Instrument, New Castle, DE, USA) with a film/fiber tension 850/800 clamp. The films were cut using a guillotine cutter to 15 × 5 mm size and loaded to the DMA applying ramp force of 1 to 18 N/min at 25 °C. Further, stress and strain data sets were processed to obtain tensile properties.

### 2.6. Fourier-Transform Infrared Spectroscopy (FT-IR)

The spectra of dried BNC films were recorded using a Perkin-Elmer Spectrum One FTIR spectrometer (Perkin Elmer Inc., Washington, DC, USA) equipped with a universal ATR sampling accessory and Perkin Elmer software (Spectrum 10™). Each spectrum was obtained by acquiring 20 scans with a spectral resolution of 4 cm^−1^ (4000–650 cm^−1^).

### 2.7. Thermogravimetric Analysis (TGA)

To assess the thermal properties of the dried BNC samples, thermogravimetric analysis was conducted using a Pyris 1 TGA instrument (PerkinElmer, Waltham, MA, USA). Each film weighing 10 mg underwent a heating process from 25 to 800 °C at a temperature ramp rate of 10 °C/min to generate thermogravimetric curves.

## 3. Results and Discussion

In this study, we conducted a comprehensive investigation into the properties of BNC derived from the strain *Komagataeibacter medellinensis* ID13488. Our study focused on evaluating the effects of downstream treatments on the structural and mechanical characteristics of the produced BNC to establish proper downstream methodology for certain type of material application.

When varying KOH concentration within the range of 0.2 to 1.0 M, we found no evidence of weight difference, which could indicate that the KOH interaction has no major effect on the water holding ability in such a range, which is different to higher concentrations [32]. On the other hand, when we conducted a screening of NaOH concentration (ranging from 0.2 to 1.0 M), a decreasing degradation curve was observed, which clearly indicates a weight variation after the drying process. This weight appears to be directly related to the NaOH concentration (Figure 1), suggesting a possible increment in cellulose chain compaction due to interactions with sodium cations [33,34], which could result in a decreased water content since the mercerized composite has a thinner interfacial matrix layer and fewer voids than the un-mercerized composite [35].

After submitting the membranes to either autoclaved or alkaline downstream processes, now at 0.1 M with in a different batch, comparing the weight means, the pooled standard deviation was 0.04, and all 95% confidence intervals (CI) overlapped with each other at least once, from the inferior range of 0.04–0.16 (KOH se) to 0.16–0.2859 (NaOH se). Therefore, it can be concluded that the means of the five groups are not significantly different from each other at an α = 0.05, indicating that tested processes did not have a major effect on the membranes final weight, regardless of the exposure time (Table 2). Upon analyzing the thickness of the BNC samples, the data revealed significant variations in the thickness of BNC membranes related to the treatment methods employed (Figure 2). In particular, BNC samples subjected to autoclaving displayed the thicker films, with a mean thickness of 3.18 × 10^−2^ mm—around 63% superior to the second highest mean of 1.94 × 10^−2^ mm from prolonged exposure treatment with KOH, while the thickness between the other alkaline-treated samples did not differ significantly. These results align with the dry weights yield, as autoclave-treated membranes seem to preserve their three-dimensional (3D) configuration, in contrast to BNC collapsed structures affected by the cationic interaction [36].

### 3.1. SEM Images

SEM analysis revealed distinct surface characteristics between NaOH-treated and KOH-treated BNC membranes compared to autoclave-treated ones. NaOH-treated BNC membranes exhibited consistent surface characteristics regardless of the duration of exposure (Figure 3). On the other hand, KOH-treated membranes displayed minor discrepancies, which, like previous findings, appeared to be not significant. However, both alkaline-treated surfaces presented noticeable deviations when compared with the autoclaved membranes. The NaOH-treated surfaces displayed a more uniform and smoother morphology, indicative of enhanced compaction of the cellulose chains. In contrast, the autoclaved membranes displayed a rugged and irregular surface, signaling a less densely packed 3D structure [19].

### 3.2. Dynamics Mechanical Analysis (DMA)

It was observed that autoclave-treated BNC exhibited superior flexibility compared to alkaline-treated BNC samples. The higher Young’s Modulus of the alkaline-treated samples indicated greater stiffness [37]. Furthermore, it was noted that there were significant variations in BNC thickness related to the treatment methods employed. Autoclave-treated membranes preserved their three-dimensional configuration, in contrast to BNC structures, which were apparently affected by cationic interactions [38] due to the mercerization treatment. It was observed that autoclave-treated BNC exhibited superior flexibility compared to alkaline-treated BNC samples. The higher Young’s Modulus of the alkaline-treated samples indicated greater stiffness [37]. Furthermore, it was noted that there were significant variations in BNC thickness related to the treatment methods employed. Autoclave-treated membranes preserved their three-dimensional configuration, in contrast to BNC structures which were apparently affected by cationic interactions [38] due to the mercerization treatment.

Mechanical properties of the BNC samples prepared via different downstream processes were evaluated by DMA measurement, and the recorded parameters were presented in Table 3. The results obtained from the tensile behavior analyses of BNC samples provided insights into the relationship between the mechanical properties and the downstream processes, as well as the influence of alkaline treatments on BNC. It was demonstrated that the treatment of the BNC samples with autoclave process and alkali solutions with the variation of time and type of compound affected their properties since traces of contaminants that were suspected to influence the formation of hydrogen bonding were removed by the treatment. 

Firstly, the autoclave-treated bacterial nanocellulose (BNC5) exhibited a Young’s Modulus of 1821.2 ± 523.8 MPa, indicating its relatively lower stiffness when compared against the alkaline-treated BNC samples. Moreover, BNC5 displayed a tensile strength of 39.1 ± 5.3 MPa and an elongation at break of 3.12 ± 0.7%. Notably, BNC5 has demonstrated superior flexibility compared to the alkaline-treated BNC samples.

These results emphasize the significant impact of treatment methods on the mechanical properties of BNC. While the alkaline-treated samples exhibited better stiffness, the autoclave-treated one contrasts by exhibiting superior flexibility, as lower Young’s modulus indicates. This information contributes to a better understanding of how different treatments can tailor BNC properties for specific applications, offering opportunities for materials which bacterial nanocellulose is the main compound, given that a thermal process can provide better handling, in contrast to the brittle property from an alkaline solution treatment [39].

### 3.3. FT-IR Spectra

The chemical characterization of the BNC samples was performed by FT-IR analysis. In general, the collected results showed that neither treatment created structural differences between produced BNC products. The main bonds in all samples can be assigned as that CH stretching of CH_2_ at 2913 cm^−1^, water OH bending at 1645 cm^−1^, CH_2_ symmetric bending at 1426 cm^−1^, CH bending at 1376 cm^−1^, C-O-C stretching at 1160 cm^−1^, antisymmetric out-of-phase bending at 1119 cm^−1^ and 1053 cm^−1^. The region 3600–3200 cm^−1^ corresponds to the hydroxyl functional groups and hydrogen bonds [40]. It was also detected that the intensity of the peaks, which can be attributed to the methyl and methine bending vibration on the backbone at 1426 cm^−1^ and 1335 cm^−1^, increased with the alkaline treatment. This result can explain the increase in physical interactions between hydroxyl groups from the bacterial cellulose structure with the alkaline condition [40]. The intensity of peaks in this region of BNC samples produced with the long exposure, 12 h, treatment with KOH and NaOH is higher than the samples obtained with the short exposure treatments, while autoclaved BNC has the lowest intensity. These results demonstrated the increased formation of hydrogen bonds as the interaction time between bacterial nanocellulose and alkaline solutions increases. This behavior is already observed when comparing the sample subjected to heat treatment against those with short-duration alkaline treatment, and it intensifies as the interaction time increases, as observed either on the long exposures to NaOH and KOH. These results also matched the mechanical properties of the produced samples mentioned above. H-bonds could increase Young’s modulus value by reducing the free volume, thus resulting in the samples having higher H-bonds having higher Young’s modulus.

The FT-IR results presented in Figure 4, as expected, clearly demonstrate no structural differences between applied treatments. In fact, what stands out the most among the spectra is the clear increased formation of hydrogen bonds as the interaction time between bacterial nanocellulose and alkaline solutions increases. This behavior was already observed when comparing the sample subjected to heat treatment against those samples subjected to short-duration alkaline treatment, and it intensifies as the interaction time increases, as observed either on the long exposures to NaOH or KOH.

Given that the alkaline interaction process tends to collapse the 3D configuration of cellulose, as confirmed in the SEM images, it is expected that the cellulose chains will approach each other and form more intermolecular interactions via hydrogen bonding, reinforcing the brittleness property indicated in previous experiments.

### 3.4. Thermogravimetric Analysis (TGA)

The thermal behaviors of the BNC membranes with different downstream treatments were investigated by TGA analysis (Figure 5). In general, the thermal degradation steps of BNC can be summarized as dehydration, depolymerization, and decomposition of glycosyl units [41]. The first main degradation step, corresponding to dehydration, depolymerization, and decomposition of the glycoside units, occurred at the temperature range between 150 and 400 °C. It was detected that the degradation behavior of the BNC treated by autoclave method showed significant differences from that of BNC samples obtained by the alkaline treatment method. It was observed that the degradation of the autoclaved product started at a lower temperature and occurred over a broader temperature range. The onset of thermal degradation temperature (at 5% weight contrast) for autoclaved BNC was observed at 155 °C, which was much lower than that in the alkaline-treated samples, which was found to be around 246 °C for almost all samples. In general, there were no significant differences between the BNC samples produced via alkaline treatment in the first step in terms of onset temperatures of the degradation behavior. However, different thermal behavior was detected between the BNC samples in the second thermal degradation step, which included oxidation and breakdown of carbonaceous residues and occurred within the temperature range of 450–650 °C.

As seen in the TGA diagram of the BNC treated with KOH, the residual mass was calculated as 10.5% and 10.8% for the samples coded as BNC3 and BNC2, respectively. These values were found to be 11.2% for the BNC4 sample treated with NaOH, while the residual mass of BNC1 and BNC5 were 11.237% and 21.94%, respectively. These results demonstrated that the BNC samples treated with a KOH alkaline environment had higher purity than the other treated BNC samples. Overall, these results showed that a substantial thermal difference was achieved through alkaline treatments. In essence, these treatments have effectively increased the material’s thermal resistance, irrespective of the type or duration of alkaline exposure, clearly illustrating the potential for obtaining distinct proprieties variations of BNC based on the modulation of the sterilization and disinfection method.

From the thermogravimetric behavior presented in Figure 5, it is observed that the BNC subjected to autoclaving initiates decomposition at around 200 °C, while BNC treated with alkaline solutions exhibits a higher thermal stability, with decomposition starting at approximately 300 °C. This demonstrates a substantial thermal difference achieved through alkaline treatments. In essence, these treatments have effectively increased the material’s thermal resistance, irrespective of the type or duration of alkaline exposure, clearly illustrating the potential for obtaining distinct propriety variations in BNC based on the modulation of the disinfection method. Importantly, these findings align with results obtained from other previous characterizations such as FT-IR and DMA.

## 4. Conclusions

In the course of our investigation, this study set out to explore the impact of distinct treatment methods on the mechanical and structural attributes of bacterial nanocellulose (BNC) derived from the *K. medellinensis* ID13488 strain. Our work employed diverse techniques, including scanning electron microscopy (SEM), Fourier transform infrared spectroscopy (FTIR), and dynamic mechanical analysis (DMA), to scrutinize the morphology, chemical composition, and mechanical behavior of BNC specimens subjected to autoclave and alkaline solutions. The findings underscore the profound influence of treatment methods on BNC properties, manifesting alterations in crystallinity, hydrogen bonding network, and nanofiber flexibility.

Notably, our investigation disclosed that autoclave-treated BNC demonstrated superior flexibility, preserving its three-dimensional configuration, while alkaline-treated BNC exhibited increased stiffness and reduced thickness. These outcomes signify the potential customization of BNC properties through diverse downstream processes, catering to specific industrial demands where high purity and distinct mechanical characteristics are imperative. Our work adds depth to the existing literature by offering a comprehensive comparative analysis of the effects of various treatment methods on BNC properties. Additionally, we introduce a novel bacterial strain, *K. medellinensis* ID13488, as a promising large-scale BNC producer.

However, our investigation acknowledges certain research limitations and challenges, including difficulties in controlling environmental factors affecting BNC production, the absence of standardized evaluation methods for BNC properties, and a scarcity of studies on its long-term stability and biodegradability. Consequently, we recommend further research to address these issues and explore potential BNC applications in biomedicine, electronics, and textiles. In conclusion, our group’s work underscores the promise of *K. medellinensis* ID13488 as a large-scale BNC producer, emphasizing the pivotal role of downstream treatments in tailoring its properties for diverse applications. Ongoing research in this direction holds significant potential for advancing BNC utilization in industries requiring specific material characteristics.

## Figures and Tables

**Figure 1 polymers-16-00678-f001:**
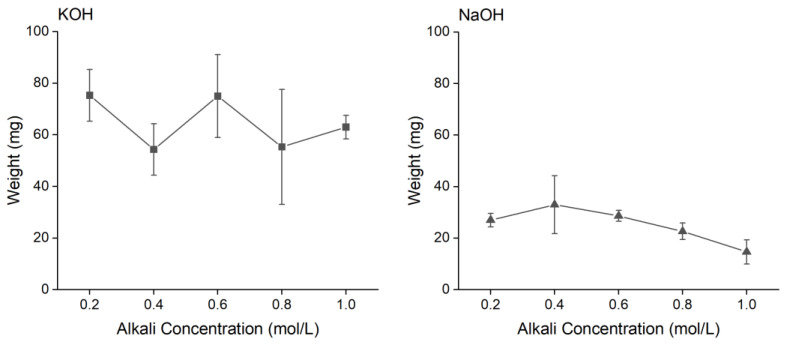
BNC membranes weight after treatments with different concentrations of NaOH and KOH. Note that the NaOH and KOH treatments were performed on separate batches of BNC membranes.

**Figure 2 polymers-16-00678-f002:**
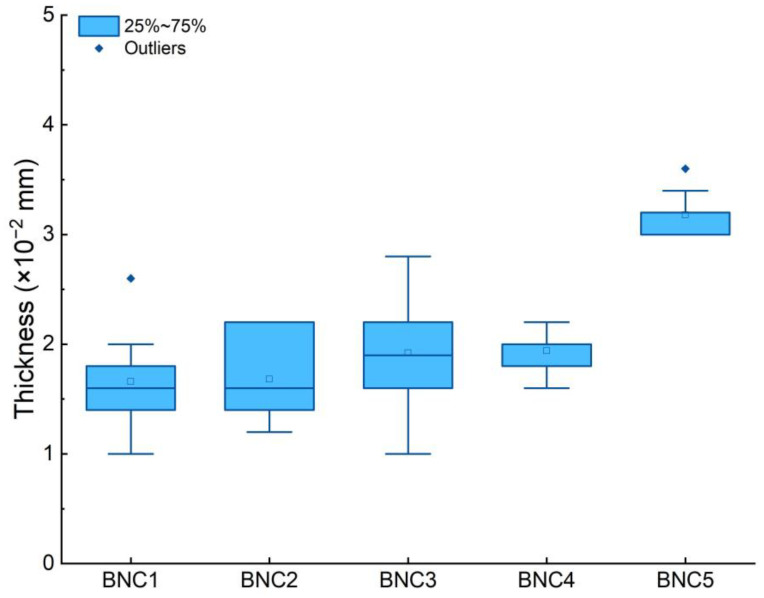
Thickness of BNC membranes cultivated from *K. medellinensis* (ID13488) for 10 days, after alkaline treatments with different exposure times and after autoclave treatment.: NaOH, short exposure (BNC1); NaOH, long exposure (BNC2); KOH, short exposure (BNC3); KOH, long exposure (BNC3); autoclave (BNC5).

**Figure 3 polymers-16-00678-f003:**
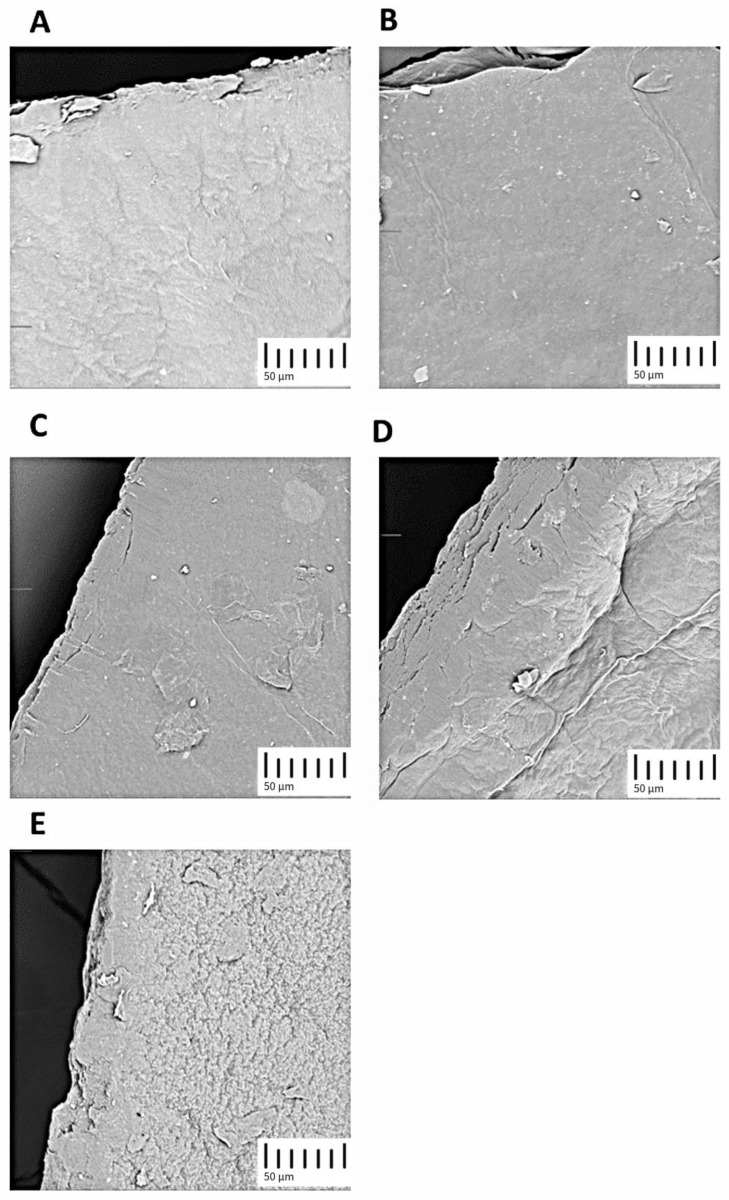
SEM micrographs of obtained BNC membranes (1kx): (**A**) NaOH, short exposure (BNC1); (**B**) NaOH, long exposure (BNC2); (**C**) KOH, short exposure (BNC3); (**D**) KOH, long exposure (BNC3); (**E**) autoclave (BNC5).

**Figure 4 polymers-16-00678-f004:**
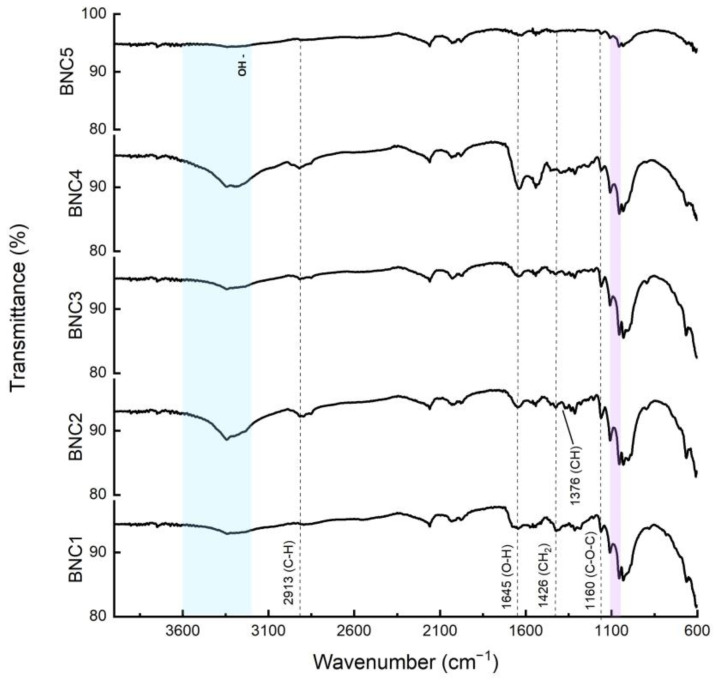
FT-IR spectra of derived BNC membranes: NaOH, short exposure (BNC1); NaOH, long exposure (BNC2); KOH, short exposure (BNC3); KOH, long exposure (BNC4); autoclave (BNC5), antisymmetric out-of-phase bending in purple and hydroxyl functional groups in blue.

**Figure 5 polymers-16-00678-f005:**
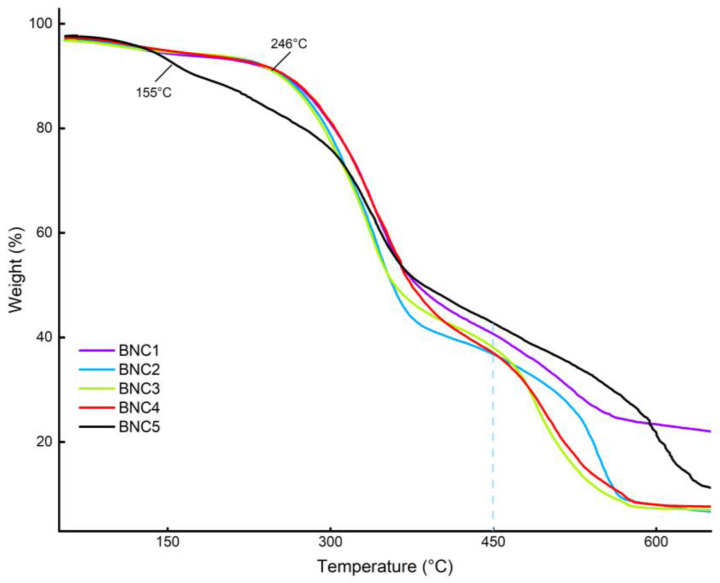
TGA analysis of obtained BNC membranes: NaOH, short exposure (BNC1); NaOH, long exposure (BNC2); KOH, short exposure (BNC3); KOH, long exposure (BNC4); autoclave (BNC5).

**Table 1 polymers-16-00678-t001:** Different treatments used for BNC downstream processing.

Sample Names	Treatment	Conditions
BNC1	NaOH Short Exposure	0.1 M NaOH for 60 min
BNC2	NaOH Long Exposure	0.1 M NaOH neutralized for 12 h
BNC3	KOH Short Exposure	0.1 M KOH for 60 min
BNC4	KOH Long Exposure	0.1 M KOH neutralized for 12 h
BNC5	Autoclave	Disinfection with steam in autoclave

**Table 2 polymers-16-00678-t002:** BNC membranes weight after treatments with equal concentrations of NaOH and KOH.

Sample	Mean (mg)	StDev (mg)	95% CI
BNC1	0.22	0.040	(0.167, 0.285)
BNC2	0.14	0.010	(0.080, 0.199)
BNC3	0.22	0.075	(0.164, 0.282)
BNC4	0.10	0.020	(0.044, 0.162)
BNC5	0.22	0.052	(0.160, 0.279)

StDev standard deviation; NaOH, short exposure (BNC1); NaOH, long exposure (BNC2); KOH, short exposure (BNC3); KOH, long exposure (BNC3); autoclave (BNC5).

**Table 3 polymers-16-00678-t003:** DMA results of BNC membranes obtained after different downstream approaches: NaOH, short exposure (BNC1); NaOH, long exposure (BNC2); KOH, short exposure (BNC3); KOH, long exposure (BNC3); autoclave (BNC5).

Sample	Young’s Modulus [MPa]	Tensile Strength [MPa]	Elongation at Break [%]
BNC1	2937 ± 683	52.2 ± 9.1	1.99 ± 0.4
BNC2	3269 ± 701	42.8 ± 6.8	1.34 ± 0.4
BNC3	2753 ± 317	45.2 ± 1.7	1.74 ± 0.1
BNC4	3263 ± 498	53.8 ± 8.5	1.93 ± 0.2
BNC5	1821 ± 523	39.1 ± 5.3	3.12 ± 0.7

## Data Availability

Data are contained within the article.

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
