# Peer review of "Targeting Bacterial Nanocellulose Properties through Tailored Downstream Techniques"

_polymers, 2024, doi:10.3390/polym16050678_

Round 1

Reviewer 1 Report

Comments and Suggestions for Authors

The presented manuscript is devoted to the production of bacterial cellulose with specified characteristics. At the very beginning of the literature review, the authors focus attention on the purity of the resulting product, its differences from wood cellulose, the difficulties of removing bacterial residues, etc. Consideration of these issues, in my opinion, is important.

The list of used literature contains only 29 titles and, in my opinion, should be expanded.

Line 46. Using 0.5M NaOH is not always sufficient to completely remove bacterial residues. The work https://doi.org/10.3390/pr8020171 indicates the use of 1 M NaOH for 3 days to remove endotoxins. Also, along with alkaline solutions, other reagents are used, for example, RIPA.

Line 89. "treasments" perhaps where the authors had to enter "treatments"

Instead of Figure 1, I would recommend placing a table including the designation of samples and the conditions for their production.

Lines 151-156. From the presented dependencies, it seems that KOH affects BC more than NaOH?!

Figure 3. What does the red dot on the graph mean? The figure caption contains designations for samples that are not used in the figure.

The quality of the spectra shown in Figure 5 does not allow for a decent comparison. The main stripes described in the manuscript are not indicated in the figure. From the obtained spectra it is necessary to extract information about the structure of cellulose. Because without this data, the work becomes meaningless.

Line 292,293. I cannot agree with this statement. So in others' work, it was shown that adding organosilicon additives to the system leads to an increase in the carbon residue.

Line 296, 297. "while the residual mass of autoclaved and NaOH s.e. were 24.7% and 26.4%, respectively." - this statement needs to be verified.

From the presented findings it follows that the background to obtaining BC will determine its properties. On the other hand, the presented manuscript overlooks the BC structure. Why don’t the authors use the X-ray diffraction method and make full use of IR spectra? Without such data, it is difficult to assess the observed differences between samples. The work is currently unfinished and cannot be published.

Comments on the Quality of English Language

Line 89. "treasments" perhaps where the authors had to enter "treatments"

Reviewer 2 Report

Comments and Suggestions for Authors

Given the potential of the strain K. medellinensis ID13488 to function as a Bacterial cellulose (BC) producer under industrially relevant and environmentally sustainable conditions, it is important to conduct a more comprehensive investigation into the properties of the material derived from this species and how downstream treatments may influence its application. To address this, the aim of this study was to optimise alkaline and autoclave treatments treated BC samples, in order to explore and compare the effects of such treatments on the structural and mechanical characteristics of the prepared materials.

The results presented in this study are interesting and helpful for researchers focusing various biomedical industrial applications. Various methods have been applied to evaluate and compare the different effects of the treatments. However, before acceptance of this paper authors need to address the following listed minor issues:

1.      Key peaks in FTIR spectra need to be labelled.

2.      In TGA curves loss of mass and T of key changes need to be added.

Reviewer 3 Report

Comments and Suggestions for Authors

The manuscript under review is devoted to studying the effect of various treatments on some properties of bacterial cellulose (BC). The topic is quite relevant, since BC is a promising material for various applications. However, the manuscript suffers from a number of significant shortcomings.

Materials and Methods: The autoclave processing is not described at all—no autoclave brand, no manufacturer’s name, no temperature and no processing time. It is unclear whether the samples being processed were in the liquid (deionized water) or vapor (steam) phase. The exact time should be specified for “long exposure”, because the vague word “overnight” is suitable for washing rather than for chemical treatment.

There is a complete discrepancy in the designations of the samples. They are designated as BC1...BC2 and as “NaOH/KOH short/long exposure”, and in different places in different ways: SE, S.e., s.e., se… It is necessary to restore order here.

There are contradictory issues. E.g., lines 98 and 101: was the alkali treatment carried out for 60 min or 2 hours? Line 110: “After sterilization...”, but was sterilization (getting rid of microorganisms) the purpose of this autoclaving? Line 240: are there −CH3 groups in cellulose?

There are few concerns about the figures. Fig. 2: such scale factors as 10−1 and 10−2 are not justified, just move the decimal point. “Weight” is plotted along the y-axis, whilst the figure caption says “weight loss”, the authors need to choose one thing. If this is really “weight loss”, then it would better to give it in relative units; but if it is simply “weight”, then the authors need to mark a point corresponding to the initial weight (or indicate it in the figure caption). The same question for Table 1: “weight” or “weight loss,” no units of measurement. And this table allows you to plot one more point in Fig. 2, at the alkali concentration of 0.1 M. By the way, why was the alkali concentration for the treatment (0.1 M) not selected from the range of previously studied concentrations (0.2–1.0 M), but was taken outside it?

Fig. 3: please indicate not only the type of treatment, but also the time for cultivating microbes, which the film’s thickness primarily depends on.

Table 2: the number of significant digits in column 2 is redundant.

Fig. 5: “Wavelength” along the x-axis is measured in cm−1, so it is really “wavenumber”.

Fig. 6: there are values 24.7% and 26.4% in the text below, but it is clear from the figure that only one curve ends above 20%, the others going below. The black curve continues to decline, so the residual weight can only be estimated from above. The authors write (lines 292 and 293): “The residual mass in the TGA curves is an important indicator on showing the purity of the produced material,” however, first, they make no attempt to analyze the dry residue (in particular, for Na and K), and second, they do not take into account the different moisture content of the initial samples. The same air drying conditions (room temperature, 48–72 h) do not at all provide the same moisture content, since different treatments would give different cellulose structures (as can be seen, in particular, from the FTIR spectra), therefore the same chemical potential of water (equal to that for water vapor in air) may be achieved at different water contents.

Line 273: «depolymerization, dehydration, and decomposition of glycosyl units»—dehydration should be placed before depolymerization, because intermolecular bonds are weaker than covalent ones.

A number of typos have been noticed. Line 46: “...range from 0.5 to 0.1 M”—this is either 0.05–0.1 M, or 0.1–0.5 M. Line 84: “BNC membranes”—there was no such abbreviation. The abbreviation “BC” is explained in lines 10, 27 and 144. Line 156: “...appears is...” Line 171: “α = 0.5”, but above it was 95%, which means α = 0.05. Line 181: “...after treatments same concentrations...”—missing “of the”. Lines 175 and 176: the superscript is not raised. Lines 243–245: “The intensity of peaks... is more intensive than...”

As an optional recommendation, moving the beginning of the “Results and Discussion” to the “Introduction” could be suggested.

The manuscript can be published after the listed shortcomings are eliminated.

Round 2

Reviewer 1 Report

Comments and Suggestions for Authors

The authors of the manuscript answered the questions posed earlier, but while re-working with the manuscript, additional questions arose.

I suggest authors expand the list of keywords. The authors use ATR-FTIR, so it is necessary to check and correct the designation on the spectrum axes. It is desirable to increase the quality of SEM micrographs. The presented micrographs indicate a dense morphology; perhaps partial dissolution of cellulose occurred with further reprecipitation of the polymer in water?

Lines 254-258. It is necessary to add a link to the above statement.
The quality of the IR spectra is still not very good.

Reviewer 3 Report

Comments and Suggestions for Authors

Fig. 1: on the ordinate axis the authors left absolute weight (instead
of the recommended relative one), but then they need to indicate the
initial weight in the caption or on the figure itself.
Designation on the abscissa axis: "Alkaline Solutions", but "Alkali
Concentration" is needed.
No units of measurement in the column headings of Table 2.

Round 3

Reviewer 1 Report

Comments and Suggestions for Authors

The authors use ATR for IR spectroscopy, while on the presented spectra along the y axis the designation Transmittance? If possible, it is better to replace SEM micrographs with higher magnification. I also recommend making the scale bar more readable.
